# Facile Solvent-Free Synthesis of Metal Thiophosphates and Their Examination as Hydrogen Evolution Electrocatalysts

**DOI:** 10.3390/molecules27165053

**Published:** 2022-08-09

**Authors:** Nathaniel Coleman, Ishanka A. Liyanage, Matthew D. Lovander, Johna Leddy, Edward G. Gillan

**Affiliations:** Department of Chemistry, University of Iowa, Iowa City, IA 52242, USA

**Keywords:** metal thiophosphates, solvent-free metathesis, electrocatalyst, hydrogen evolution reaction

## Abstract

The facile solvent-free synthesis of several known metal thiophosphates was accomplished by a chemical exchange reaction between anhydrous metal chlorides and elemental phosphorus with sulfur, or combinations of phosphorus with molecular P_2_S_5_ at moderate 500 °C temperatures. The crystalline products obtained from this synthetic approach include MPS_3_ (M = Fe, Co, Ni) and Cu_3_PS_4_. The successful reactions benefit from thermochemically favorable PCl_3_ elimination. This solvent-free route performed at moderate temperatures leads to mixed anion products with complex heteroatomic anions, such as P_2_S_6_^4−^. The MPS_3_ phases are thermally metastable relative to the thermodynamically preferred separate MP_x_/ MS_y_ and more metal-rich MP_x_S_y_ phases. The micrometer-sized M-P-S products exhibit room-temperature optical and magnetic properties consistent with isolated metal ion structural arrangements and semiconducting band gaps. The MPS_3_ materials were examined as electrocatalysts in hydrogen evolution reactions (HER) under acidic conditions. In terms of HER activity at lower applied potentials, the MPS_3_ materials show the trend of Co > Ni >> Fe. Extended time constant potential HER experiments show reasonable HER stability of ionic and semiconducting MPS_3_ (M = Co, Ni) structures under acidic reducing conditions.

## 1. Introduction

Hydrogen is an important fossil fuel alternative with high energy density (~120 kJ/g) and potentially low environmental footprint [1,2]. There is strong interest in the electrochemical splitting of water into hydrogen and oxygen using renewable energy resources (green hydrogen). Sluggish kinetics for electrochemical water splitting, particularly the oxygen evolution reaction (OER), limits effective large-scale hydrogen production [3]. Expensive precious-metal catalysts, such as RuO_2_ and IrO_2_ for OER [4,5], and Pt for the hydrogen evolution reaction (HER) [6,7], show high activity for electrocatalytic water splitting, though in some cases they suffer from long-term stability issues. In addition to complex metal oxides, there is renewed interest in investigating non-oxide materials with surface structures that can catalyze water splitting electrochemistry. A diverse range of transition-metal phosphides, nitrides, sulfides, oxides, carbides, and borides reportedly show electrocatalytic activity for water splitting reactions [8,9,10,11,12,13,14].

Within the M-P and M-S families, several compositions and structures show electrocatalytic HER activity; these include Ni_2_P, CoP, MoS_2_, CoS_2_, and NiS_2_ with appreciable electrocatalytic activity and stability in electrochemically reducing environments in acidic electrolytes [14,15,16,17]. Our recent synthetic and electrocatalytic work describes solvent-free direct MCl_2_/P reactions to form phosphorus-rich MP_2_ and MP_3_ materials that have moderate HER catalytic behavior and robust stability in acid [18]. The most HER active phosphides, i.e., those with the lowest applied potentials, were CoP_3_ and NiP_2_. Anion-rich structures have lower metal content and potentially protective polyanion networks that may better shield the metals from degradation in corrosive environments versus metal-rich systems. While phosphides and sulfides are frequently grown as nanostructures on porous supports, well-defined syntheses of micrometer-sized products are desirable to examine bulk properties and minimize size-dependent effects on catalytic behavior. Theoretical studies on metal phosphides indicate that surface P-H bonding may be more favorable for HER kinetics and so non-metal anions on the solid surface may be as important or more important than surface transition metal cations [14,19,20,21,22,23].

In addition to anion-rich metal phosphide or sulfide structures, there exist many non-metal-rich thiophosphate structures that contain P_x_S_y_ anions in extended ionic solid networks. Solid-state materials in the layered MPS_3_ family contain P_2_S_6_^4−^ anions, are often semiconducting [24,25], and show isolated metal-ion paramagnetism [26,27]. In addition to MPS_3_ (M= Mn, Fe, Co, Ni), other observed crystalline M-P-S phases include those closer to MX_2_ compositions (MP_x_S_2-x_ such as CoP_0.5_S_1.5_) [28]. Layered MPS_3_ structures have been extensively studied for their electronic, magnetic, and intercalative properties [29,30,31,32,33]. These anion-rich layered structures have been examined as materials for the reversible intercalation of lithium and sodium ions in battery studies [34,35,36,37,38]. The MPS_3_ materials have also been studied as photo-electrocatalysts and hydrogen storage materials [39,40,41]. There are recent studies of MPS_3_ materials as heterogeneous electrocatalysts for hydrogen reduction and oxygen evolution reactions (HER and OER) [42,43,44,45]. Several recent comprehensive reviews compare and contrast MPS_3_ structures, electronic and catalytic properties, and photochemical behavior, which serves to highlight resurgent interest in these compounds with catalytic activity and low metal content [46,47,48].

The syntheses of MPS_3_ materials often require careful and extended heating to produce these P/S-rich structures without decomposing them into other more stable metal-rich structures or poorly crystalline products. For example, MPS_3_ structures are typically produced by heating pure elements at ~650 °C for 5–17 days [38,42], or at 750 °C for 7 days [45]. Partially crystalline MPS_3_ materials are grown from layered metal oxides reacted with phosphorus and sulfur at 520 °C [37]. Crystalline MPS_3_ is also produced from a reactive P_2_S_5_ melt with elemental metal powders heated at 580–650 °C for 2–5 days [49]. The P_2_S_5_ reaction study noted that 580 °C was the maximum CoPS_3_ reaction temperature to avoid formation of CoPS impurities. Overheating the products of MPS_3_ reactions may lead to the decomposition into multi-phase products, including binary phosphides, sulfides, and lower content thiophosphates. The single-step, solvent-free, moderate temperature synthesis of crystalline MPS_3_ materials is challenging.

This report describes facile exchange reactions that are solvent-free and thermochemically driven to produce materials in the M-P-S family via reactions of anhydrous metal chlorides with phosphorus and either sulfur or pre-bonded molecular P_2_S_5_ at 500 °C. The successful reactions are explained in the context of reaction thermochemistry. The structure, composition, and physical properties of the M-P-S products are reported. Crystalline single-phase MPS_3_ (Fe, Co, Ni) materials synthesized by this solvent-free exchange reaction are examined as HER electrocatalysts in 0.5 M H_2_SO_4_ and compared with prior electrocatalytic results for these thiophosphate materials.

## 2. Results and Discussion

### 2.1. M-P-S Synthesis from Elemental Phosphorus and Sulfur

Initial reactions between metal chlorides and combinations of elemental phosphorus and sulfur (P + S) showed that MPS_3_ phases or Cu_3_PS_4_ form in these heated solvent-free exchange reactions, so these phases were stoichiometrically targeted along with a thermodynamically stable PCl_3_ byproduct. The M-P-S reactions using P/S elemental reactants for either MPS_3_ or Cu_3_P_4_ products are shown in Equations (1) and (2). In each case, the P/S reactant is stoichiometrically balanced to match the target phase and provide excess phosphorus to sequester the chlorine as volatile PCl_3_ (mp −94 °C, bp 76 °C).
3 MCl_x_ + (3 + x) P +9 S → 3 **MPS_3_** + x PCl_3_ (x = 2 for Ni, Co, x = 3 for Fe)(1)
3 CuCl_2_ + 3 P + 4 S → **Cu_3_PS_4_** + 2 PCl_3_(2)

The chosen metal dichlorides have melting points above the 500 °C reaction temperature (~700–1000 °C), while FeCl_3_ melts and decomposes to FeCl_2_ around 310 °C that melts at 667 °C. The non-metal reactants show low temperature phase transitions, specifically for molecular sulfur S_8_ (mp 115 °C, bp 445 °C) and polymeric red phosphorus (sublimes as P_4_ near 420 °C). At the end of the 500 °C reaction, the M-P-S products are isolated by transporting the PCl_3_ byproduct and the unreacted non-metal intermediates (if present) to the empty end of the ampoule using a temperature gradient. While sulfur chlorides such as SCl_2_ (ΔH_f_ = −50 kJ/mol) may potentially form as reaction byproducts, they have lower thermochemical stability than PCl_3_ (ΔH_f_ = −320 kJ/mol) [50]. Thiophosphate product formation via this solvent-free exchange reaction follows our prior success with phosphorus-rich MP_2_ and MP_3_ materials grown using a thermochemically driven PCl_3_ elimination strategy [18]. The thermochemical driving forces that lead to successful reactions between MCl_x_ and different P and S reactants will be described later. The naming scheme for these products from reactions using P and S is M_x_P_y_S_z_ [P + S], such as CoPS_3_ [P + S].

The M-P-S products that resulted from reactions of iron, cobalt, and nickel chlorides with elemental P/S reactants are black solids, while the Cu-P-S product is a green solid. Powder X-ray diffraction (XRD) analysis of these metal thiophosphate products show that several of these products form crystalline MPS_3_ structures (Figure 1). The major phases that are observed include monoclinic FePS_3_ (PDF #00-033-0672) and monoclinic NiPS_3_ (PDF #01-078-0499) with several small peaks for cubic NiS_2_ (PDF #04-003-4307). In the case of Co-P-S, two similar intensity crystalline phases are identified by XRD, monoclinic CoPS_3_ (PDF #01-078-0498), and cubic-phase CoP_0.5_S_1.5_ (Co_2_PS_3_, PDF #04-007-4518). This latter phase adopts the pyrite CoS_2_ structure with phosphorus substituting some sulfur in a solid-solution formula of CoP_x_S_2-x_ where x = ~0.50 [28]. The XRD pattern for the Cu-P-S shows only crystalline peaks for orthorhombic Cu_3_PS_4_ (PDF #04-004-0447).

### 2.2. M-P-S Synthesis from Elemental P and P_2_S_5_

In a similar fashion to the elemental reactions described above, MPS_3_ and Cu_3_PS_4_ reactions were performed using a molecular P_2_S_5_ reactant (mp 288 °C, bp 514 °C) containing pre-bonded phosphorus and sulfur atoms. Additional red phosphorus was used to target the desired M-P-S product with PCl_3_ byproduct formation (Equations (3) and (4)).
15 MCl_x_ + 9 P_2_S_5_ + (5x−3) P → 15 **MPS_3_** + 5x PCl_3_ (x = 2 for Ni, Co, x = 3 for Fe)(3)
15 CuCl_2_ + 4 P_2_S_5_ + 7 P → 5 **Cu_3_PS_4_** + 10 PCl_3_(4)

The XRD patterns of the products synthesized from the metal halides and P_2_S_5_/P are shown in Figure 2. Like the elemental reactions, the metal thiophosphates crystallized as monoclinic FePS_3_, monoclinic CoPS_3_, and monoclinic NiPS_3_, and orthorhombic Cu_3_PS_4_. The P_2_S_5_ reaction with cobalt chloride produced single-phase CoPS_3_ versus the elemental P/S reaction. No impurity phases were identified in the XRD patterns, though several peaks have different relative intensities as compared with their standard patterns, which may indicate preferred growth along a crystallographic direction or stoichiometry differences. The naming scheme for these products from reactions using P_2_S_5_ and P is M_x_P_y_S_z_ [P_2_S_5_ + P], such as CoPS_3_ [P_2_S_5_ + P].

The unit cell structures of the MPS_3_ and Cu_3_PS_4_ products are shown in Figure 3. Extended structures are shown to illustrate the layering of the MPS_3_ materials with anion-rich layers nearest to the interlayer region. The MPS_3_ structures contain 2 M^2+^ cations with each P_2_S_6_^4−^ anion, which form PS_3_ pyramidal units that point into the interlayer spaces above/below the metal containing layer. The MPS_3_ form isostructural monoclinic structures with all three lattice parameters shrinking from Fe to Co to Ni in the MPS_3_ structure (see Appendix A for details). In contrast, Cu_3_PS_4_ is an orthorhombic three-dimensional structure that contains three Cu^+^ cations and PS_4_^3−^ anions.

### 2.3. Compositional Analysis of M-P-S Products

Table 1 summarizes key XRD and product yield information for the elemental P/S and P_2_S_5_/P reaction products targeting MPS_3_ (M = Fe, Co, Ni). The product yields for samples range from the ~60–95% range with higher yields for phase-pure products from P_2_S_5_ reactions. Energy dispersive spectroscopy (EDS) analysis performed on the M-P-S products shows low chlorine residues for FePS_3_ from the elemental P/S reaction to nearly undetectable chlorine (<2%) for other samples, indicating effective PCl_3_ elimination, consistent with our prior work on crystalline MP_2_/MP_3_ synthesis [18]. The P/S content from EDS data is slightly higher than the ideal 1:3 ratio for MPS_3_ and the overall higher non-metal content is detected, which could indicate some excess P and/or S polymeric components on the particle surfaces. The Cu_3_PS_4_ reactions using P/S or P_2_S_5_/P reactants had high product yields (>90%) and undetectable chlorine by EDS. Semiquantitative EDS analysis translated to the relative product compositions for the (P + S) reaction of Cu_3_P_1.38_S_3.75_ and the (P_2_S_5_ + P) reaction of Cu_3_P_1.20_S_3.60_, which are near the Cu_3_PS_4_ target composition. Since EDS provides an estimate of the relative composition, it may be biased towards particle surface coatings. A more quantitative bulk analysis of the entire MPS_3_ samples was performed by ICP-OES analysis (Table 1). The bulk ICP compositional data provide good agreement with MPS_3_ formulations and the (P + S) mixed phase cobalt product has a lower P/S content consistent with its detected Co_0.5_P_0.5_S_1.5_ composition.

### 2.4. Particulate Morphologies of M-P-S Materials

Representative SEM images of the M-P-S products from the direct solvent-free exchange reaction of metal chlorides with P/S or P_2_S_5_/P reactants are shown in Figure 4 and Figure 5, respectively. The FePS_3_ samples grow as flat plate-like structures (~3–15 µm wide by 1–2 µm thick) that are mixed with microparticle aggregates. Similarly, the CoPS_3_ products form as large aggregates comprised of faceted blocky particles ~1–5 µm in size. In some cases, large monoliths consist of smaller fused octahedral particles that are several hundred nm in size. The NiPS_3_ samples consist of large aggregates of small irregular fused particles that are in the ~1–10 µm range, but with a wider range of sizes and less well-formed faceted particles than the Fe and Co samples. The Cu_3_PS_4_ samples consisted of small particle aggregates (~1–5 µm) and larger faceted monoliths (~10–20 µm). Overall, the products from P/S or P_2_S_5_/P reactions are similar in morphology, but there are indications that the P_2_S_5_ reactions may produce larger faceted particles, in the CoPS_3_ case (Figure 5). Several higher magnification TEM images of the P_2_S_5_/P MPS_3_ products are shown in Figure 6 for smaller suspended portions of these catalyst samples. The morphologies are generally small few micrometer-sized faceted particles that are fused into larger aggregates. Some of the smallest particles are near 0.5 μm in size.

### 2.5. Spectroscopic and Magnetic Analysis of M-P-S Products

The MPS_3_ (M = Fe, Co, Ni) and Cu_3_PS_4_ samples were analyzed by IR spectroscopy to identify P-S bond vibrations for the thiolate anions. The IR spectra for several reaction products are shown in Appendix A. Each sample shows a clear peak around 570–580 cm^−1^, which is characteristic of the PS_3_ asymmetric stretching vibration [41]. The P-S thiolate vibrations for FePS_3_, CoPS_3_, and NiPS_3_ from P/S reactants are at 572 cm^−^^1^, 580 cm^−1^, and 572 cm^−1^, respectively. The Cu_3_PS_4_ sample shows a similar, but less intense peak near 500 cm^−1^ due to the structural differences between the MPS_3_ and Cu_3_PS_4_. The IR data for the samples synthesized from P/P_2_S_5_ reactants displayed similar IR results.

The diffuse reflectance optical absorption results for the M-P-S materials are shown in Appendix A. All MPS_3_ samples are black, visibly reflective, and thus show broad absorption across the visible light region and do not show band gap onsets until the edge of the visible/IR region (E_g_ ~1.3–1.7 eV, 740–940 nm). The Cu_3_PS_4_ samples are brownish-green and have slightly higher band gaps and visible absorption onsets (E_g_ = 2.36 eV, ~525 nm). These optical absorptions and band gaps of single-phase products are consistent with literature reports for these MPS_3_ and Cu_3_PS_4_ materials. In contrast to observed optical properties, theoretical band structures predict a small band gap for FePS_3_ and zero band gap for NiPS_3_ and CoPS_3_ [51].

Room temperature magnetic susceptibility measurements on the M-P-S materials from either synthetic method result in magnetic moments that are generally consistent with paramagnetic spin-only magnetic moments for isolated metal ions observed for metal thiophosphates (Appendix A) [42]. Specifically, the MPS_3_ structures consist of 2 M^2+^ cations and P_2_S_6_^4−^ anions, while the Cu_3_PS_4_ structure consists of 3 Cu^+^ and PS_4_^3−^ ions. The FePS_3_ magnetic moments are consistent with d^6^ high spin Fe^2+^ (4.90 BM), while CoPS_3_ and NiPS_3_ magnetic moments are somewhat lower than those expected for d^7^ high spin Co^2+^ (3.87 BM) and d^8^ Ni^2+^ (2.83 BM), possibly due to NiS_2_ or P_x_S_y_ presence. The Cu_3_PS_4_ samples showed nearly diamagnetic properties, consistent with d^10^ Cu^+^ ions.

### 2.6. Thermochemical Comparison of MPS_3_ Materials to MP_x_ and MS_y_ Counterparts

The direct reactions of elements often require relatively high temperatures, multi-day reaction times, and produce thermodynamically stable products. In the case of elemental reactions with 3d metals and phosphorus or sulfur, the MP_x_ and MS_y_ or MX_2_, where X is a combination of P/S, products are generally more thermodynamically stable, and therefore they are more easily produced compared to the mixed anion-layered MPS_3_ structures with P_2_S_6_^4−^ dumbbell anions. A comparison of standard heats of formation [50,52], and the calculated energy/atom values [51], for several related MP_2_/MP_3_, MS_2_, MPS, and MPS_3_ products are shown in Table 2. Since the heats of formation for the mixed M-P-S phases have not been experimentally determined, comparisons are made using the calculated eV/atom formation energies. The eV/atom energy stability comparisons show that the MS_2_ structures (energies in bold) are stable relative to both the MP_x_ and MPS_3_ structures. In the cobalt case, the CoPS structure is near CoS_2_ in stability. Overall, the per atom formation energies indicate that all three MPS_3_ materials are predicted to be unstable versus MPS and M_x_S_y_ structures and P_x_S_y_ products [51].

The thermochemical stability differences are directly applicable to the elemental syntheses, which are the most common routes to synthesize bulk phosphide, sulfide, and MPS_3_ materials. Typically, complex M-P-S materials, such as MPS_3_ (M = Fe, Ni, Co) are produced via elemental reactions at temperatures of 650 °C or higher that take several days to over a week to complete reactions [42,48,53,54,55]. It has been noted that some elemental syntheses are complicated by the formation of more stable metal sulfides, such as CoS_2_. Tuning reaction thermochemistry through judicious choices of reactants and thermodynamically favored byproducts can enable reactions to proceed in a more facile manner than observed in solid–solid or solid–gas elemental reactions.

In our previous work on the solvent-free synthesis of MP_2_ and MP_3_ materials from direct metal halide reactions with elemental red P at 500 °C, the thermochemical driving force was primarily the formation of a stable PCl_3_ byproduct (ΔH_f_ = −320 kJ/mol) [18]. This led to moderately exothermic reaction enthalpies (ΔH_rxn_ from ~−20 to −100 kJ/mol); endothermic MP_x_ reactions (e.g., MnP_4_) were unsuccessful. Relevant to this current synthetic work, is whether reactions of metal halides with elemental sulfur would yield S_2_Cl_2_ (ΔH_f_ = −58 kJ/mol) and stable metal sulfides (ΔH_f_ ~150–300 kJ/mol). All such reactions for Fe, Ni, and Co are predicted to be highly endothermic (>+140 kJ/mol). These control reactions were performed, and no metal sulfides were produced, instead only metal halides and transported sulfur were detected (Appendix A).

In contrast, when both P and S (or P_2_S_5_ and P) reactants were used, the PCl_3_ elimination pathway is again thermochemically available (as shown in Equations (1)–(4)), and metal thiophosphates form in moderate to high yields. It is possible that initial M-P formation occurs between MCl_x_ and P_4_ vapor, followed by sulfur incorporation from S_8_ vapor. It is also likely that in the P/S and P_2_S_5_/P reactions with metal halides, the formation of volatile P_x_S_y_ molecular intermediates favor the production of P_2_S_6_^4−^ anions (and MPS_3_ formation) during solid–gas exchange reactions at the solid MCl_x_ surface. Molecular P_2_S_5_ and other P_x_S_y_ products are formed from heating phosphorus and sulfur [56].

All non-metal reactants used in these solvent-free M-P-S syntheses should be in the liquid or vapor state prior to the 500 °C reaction temperature being reached: sulfur S_8_ (mp 115 °C, 445 °C), polymeric red phosphorus (sublimes as P_4_ near 420 °C), and molecular P_2_S_5_ (mp 288 °C, bp 514 °C). In practice, some surface reactions are observed by ~350 °C, thus reactions on the ground MCl_x_ powder surface may initially occur with liquid sulfur or P_2_S_5_ and vaporized P_4_. Since the strategy for these thermochemically-driven chemical exchange reactions are successful for the synthesis of MPS_3_ (M = Fe, Ni, Co) and Cu_3_PS_4_ structures, it may be a synthetically useful strategy to access other metal thiophosphate and mixed metal thiophosphate structures.

### 2.7. Examination of Electrocatalytic HER Activity for MPS_3_ Products

Our prior work on MP_2_ and MP_3_ compounds, showed that the FeP_2_, NiP_2_, and CoP_3_ exhibit HER activity despite their high non-metal phosphorus content, with the order of increasing activity of CoP_3_> NiP_2_ >> FeP_2_ [18]. These results indicate that non-metal rich materials can be useful HER catalysts, despite a low metal content. Carbon wax electrodes developed in our prior work were embedded with M-P-S powders via direct adhesion to the conducting wax surface and they were examined in 0.5 M H_2_SO_4_ using linear sweep voltammetry (LSV) and constant potential time base measurements (chronoamperometry, CA). The Cu_3_PS_4_ samples showed rapid degradation or dissolution during initial HER experiments and were not studied further. The approximate thickness of the embedded MPS_3_ powders is about 40 μm, assuming 1 mg amounts and ~3 g/cm^3^ density. This estimation is likely an ideal upper limit that assumes a uniform, dense coating, but in reality, these micrometer-sized powders are loosely packed and partly embedded powders in the conducting wax surface. The single phase MPS_3_ products from P_2_S_5_ reactions, show parallels to the P-rich metal phosphides, with CoPS_3_ showing higher activity than NiPS_3_, and FePS_3_ showing low to negligible activity relative to the blank C_wax_ tip (Figure 7). A 10%Pt/C powder was also used for comparison purposes and was analyzed in the same manner as the MPS_3_ samples. The data in Table 3 show a summary of the average LSV results. The individual MPS_3_ LSV data are shown with and without an applied 85% iR correction in Appendix A and in Table 3. Both results are shown as several studies caution that care should be taken in making iR corrections on catalytic materials as they can mask catalyst charge transfer differences [57,58,59]. As expected, iR correction lowers the potentials necessary to achieve a current density of 10 mA/cm^2^ by ~50 mV and FePS_3_ shows the highest cell resistance (~350 Ω vs. ~65 Ω for NiPS_3_ and CoPS_3_). Tafel slopes for initial current flow are tabulated with representative graphs in Appendix A. Electrochemically active surface area (ECSA) data show that these large MPS_3_ crystallites have relatively low surface charge accumulation relative to the Pt/C standard powder (Table 3, Appendix A).

The LSV experiments to determine the HER activity of the MPS_3_ samples were initially performed with a platinum counter electrode (CE) and the resulting data were comparable to that shown in Table 3 (and Appendix A) using a graphite counter electrode with stable activity from CoPS_3_ and slow degradation in activity for NiPS_3_ and FePS_3_ after cycling to −750 mV RHE. These HER LSV results showed no observed Pt interference, similar to our prior MP_2_/MP_3_ studies [18]. In contrast, extended time 18 h constant potential CA measurements conducted to examine MPS_3_ stability do show an apparent Pt effect in certain cases. The 18 h CA experiments with CoPS_3_ showed little or no difference when a platinum or graphite CE was used. The higher applied potential for the CA experiments performed using a Pt CE with NiPS_3_ (−420 mV) and FePS_3_ (−750 mV) catalysts displayed a significant increase in catalytic activity of about 300% over the 18 h constant potential period (Appendix A). In contrast, when the graphite CE was used, NiPS_3_ and FePS_3_ showed nearly constant activity or slowly decreasing activity (Appendix A and Table 3), which are better representations of their HER activity and stability over time, without possible platinum deposition issues. Microprobe analysis of the NiPS_3_ and FePS_3_ electrode tips after CA experiments using the platinum CE shows M/P/S elements, but Pt, if present, is below detection limits (Appendix A).

The results of the platinum CE track well with other reports describing the possibility of platinum ion migration from platinum counter electrodes that can impact metal catalyst stability in fuel cells and apparent catalyst activity in HER, particularly under acidic conditions and in the presence of oxygen-rich environments [60,61,62,63]. Graphite working electrodes show a significant increase in HER activity over time when using a platinum CE [64]. Platinum deposits forming at the cathode arise from reduction of platinum ions in the solution produced when a platinum anode is oxidized. The time lag for the deposition of platinum on the cathode is set by the slow rate of platinum anode oxidation and low concentrations of platinum ions in solution. Close proximity of the cathode and anode with rapid stirring of the electrolyte favor platinum deposition on the cathode. While Pt^2+^ and Pt^4+^ reduction to metal occurs near −1 V vs. NHE (similar to RHE at pH~0 conditions), there are indications that these values are substantially decreased depending on the nature of bound ligands on dissolved ions; for example, the standard potential for the reduction of ligated PtX_4_^2−^ (aq) + 2e ⇌ Pt (0) + 4X- (aq) varies with the halide as 0.758 V, 0.698 V, and 0.400 V vs NHE for X- of chloride, bromide, and iodide [65]. For Pt (X)_4_^2−^ metal reduction, potentials move to lower negative potentials as the ligand X becomes larger and softer (e.g., I^−^ or SCN^−^). While the applied potentials here are below the ~1 V needed for platinum reduction, the MPS_3_ surfaces display significant bonded sulfur atoms that may serve to bind platinum ions analogous to a soft donor ligand and enhance their reduction. It is possible that dissolved P_x_S_y_ ions from the catalyst could also enhance solution transport of platinum ions. In brief, the CoPS_3_ is likely stable during HER because CoPS_3_ catalyzes H_2_ evolution at potentials positive for platinum deposition. Because NiPS_3_ and FePS_3_ reduce hydrogen at more negative applied potentials, their CA conditions require holding the catalyst at a higher applied potential that may be sufficient to more readily reduce platinum ions on the catalyst surface over long 18 h time periods when a Pt CE is used. Also, note that the standard Pt/C powder sample shows relatively low stability in acid under 18 h CA conditions (Table 3).

In addition to positioning crystalline catalyst particles in direct contact with the electrolyte solution, the carbon wax working electrode design allows for post-electrochemical analysis of the bulk catalyst by XRD and SEM in ways that are often difficult to achieve with a Nafion-embedded catalyst. Slices were taken off of the end of the wax electrode, i.e., the wax with embedded catalyst on the surface, after CA experiments and the slices were examined by powder XRD, similar to that reported in our recent metal boride paper [66]. These XRD results show clear retention of the original MPS_3_ catalyst structures in the bulk material on the electrode surface (Figure 8). While the CA experiments suggest that NiPS_3_ and FePS_3_ exhibit some loss in catalytic activity with extended hydrogen reduction activity, their bulk structures remain intact. Surface decomposition reactions or their semiconductor band gaps may impact HER activities. SEM images of the MPS_3_ solids on the wax tip after the CA experiments show generally smaller particle morphologies to those in Figure 5 (Appendix A). The CoPS_3_ and NiPS_3_ are small <1 μm sized aggregates while the FePS_3_ still shows some larger multi-micrometer sized faceted particles. Qualitative EDS elemental maps show that M-P-S elements are present in nearly MPS_3_ composition on the surface along with oxygen (Appendix A). While these MPS_3_ samples show a range of SEM and TEM particle sizes and shapes and aggregation, their ECSA values are fairly similar (Table 3). The CoPS_3_ sample shows some surface roughening during HER, but its overall activity remains stable over the 18 h CA experiment period. Despite CoPS_3_ having large crystallite sizes, its relatively larger ECSA (3x greater than NiPS_3_) may reflect its better ability to transfer charge to bound H^+^ versus NiPS_3_ or FePS_3_, which both have smaller ECSA values. It is possible that the edge facets of these MPS_3_ crystallites provide favorable sites for proton bonding.

Previous literature shows a variety of HER activities for MPS_3_ samples depending on preparation and level of catalyst support. Our HER results for crystalline unsupported MPS_3_ powders embedded on a sticky conducting carbon wax electrode in acidic electrolyte are comparable with some prior studies on bulk and some nanostructured and supported MPS_3_ materials. In basic solution, NiPS_3_ requires near −350 mV applied potentials for 10 mA/cm^2^ current densities, which is similar to our acid results [67]. Carbon nanosheet supported NiPS_3_ and CoPS_3_ show 10 mA/cm^2^ current densities in basic electrolyte near −400 and −200 mV, respectively, reflecting the higher activity for CoPS_3_, analogous to our results [68]. In contrast, other studies report CoPS_3_ and NiPS_3_ with similar HER 10 mA/cm^2^ current densities near −600 mV applied potentials in acidic electrolytes and show that CoPS_3_ is stable upon extended cycling, while NiPS_3_ is not [42]. A study of NiPS_3_ supported on graphene achieved 10 mA/cm^2^ activity near −100 mV applied potentials in acidic or basic electrolytes [69]. This study also reports −179 mV applied potential needed for unsupported NiPS_3_ to achieve a 10 mA/cm^2^ current density, similar to other studies [70]. NiPS_3_ was also shown as stable in acidic environments when cycling was limited to −400 mV. FePS_3_ materials reportedly show much lower activity than NiPS_3_ materials, consistent with our results [70]. There are several studies showing that cobalt doping into NiPS_3_ and FePS_3_ structures greatly improves HER activity [71,72], and exfoliation of layered FePS_3_ similarly improves activity [73]. Overall, the HER catalytic behavior demonstrated by the crystalline free-standing MPS_3_ materials in this work is comparable to prior studies on unsupported and supported MPS_3_ catalysts. Given that all three MPS_3_ materials have near IR band gaps (~1.5 eV) and very non-metal rich compositions, it is not surprising that their HER activity requires substantial applied potentials. It is impressive that CoPS_3_ converts H^+^ to H_2_ at relatively low applied potentials near −200 mV and exhibits good extended electrocatalytic stability towards HER. Photo-assisted electrocatalysis may further improve this material’s HER activity.

## 3. Materials and Methods

### 3.1. Starting Materials

Transition metal thiophosphates were synthesized using sealed Pyrex glass ampoules (I.D. ~9 mm, O.D. ~13 mm). The starting reactants and their respective purities are the following: FeCl_3_ (Alfa Aesar, Tewksbury, MA USA, 98%), CoCl_2_ (Alfa Aesar, Tewksbury, MA USA, 99.7%), NiCl_2_ (Alfa Aesar, Tewksbury, MA USA, 99%), CuCl_2_ (Alfa Aesar, Tewksbury, MA USA, 98%), red phosphorus (P, Aldrich, 99%), sulfur (S, Alfa Aesar, Tewksbury, MA USA, 99.5%), P_2_S_5_ (Sigma Aldrich, St. Louis, MO USA, 99%), 0.5 M H_2_SO_4_ (Fisher Scientific, Waltham, MA USA, 95–98%, 18 M diluted with 18 MΩ ultra-pure water), and 10% Pt on Vulcan XC-72 carbon (C1–10 fuel cell grade, E-Tek).

### 3.2. Synthesis of M-P-S Materials in Sealed Ampoules

Transition-metal thiophosphate synthesis was performed using anhydrous metal halides and elemental non-metal reactant mixtures (P + S) or element/molecule combinations (P_2_S_5_ + P). This work builds on our prior MP_2_/MP_3_ synthesis using direct thermal reactions between metal halides and red phosphorus [18]. Both experimental routes described in this paper were balanced to target MPS_3_ products where M = Fe, Co, Ni, except in the case of Cu-P-S reactions with CuCl_2_ that were balanced to produce Cu_3_PS_4_ as this phase was consistently formed during initial survey reactions. Typically, 0.500 g of the anhydrous metal halides FeCl_3_ (3.08 mmol), CoCl_2_ (3.85 mmol), NiCl_2_ (3.86 mmol), or CuCl_2_ (3.72 mmol) were ground with stoichiometric amounts of red phosphorus and elemental sulfur (or P_2_S_5_) using an agate mortar and pestle in the argon-filled glove box. All reactions were designed to yield the respective M-P-S product and a PCl_3_ byproduct, consistent with our prior MP_2_/MP_3_ work [18,74]. The powders were loaded in a medium-wall Pyrex glass ampoule (9 mm OD) that was then closed with a valve and Cajon fitting, and removed from the glove box. The ampoule was evacuated on a Schlenk line for ~15 min, and then flame-sealed under dynamic vacuum. All reactions were heated to 500 °C using a ramp rate of 100 °C/hr and held at 500 °C for ~18–24 h. After heating, the ampoule end without the solid was pulled out of the furnace and cooled to room temperature. A colorless liquid and a yellowish-white solid was transported to the cooler end of the ampoule. When no further volatile transport was observed, the furnace was turned off and the tube was allowed to cool to room temperature. The glass ampoule was then opened in air to separate the product from transport. The isolated solid M-P-S product was weighed and stored in an argon-filled glove box. Little or no visible reaction was observed from the transport (e.g., smoking upon reaction with air), suggesting that PCl_3_ or unreacted P_4_ was sequestered in a more air-stable form with other P_x_S_y_ transports. In air, phosphorus oxides can form from pure liquid PCl_3_ visible as fuming white cloud and from P_4_ visible as flame formation.

### 3.3. Sample Characterization

Powder X-ray diffraction (XRD) was performed on ground powders affixed to glass slides using a Bruker D8 Advance DaVinci diffractometer with nickel-filtered Cu Kα X-ray irradiation (40 kV, 40 mA, 10–80 degrees 2θ, 0.05°/step). Reference XRD patterns were generated from crystal structure data using the PowderCell program [75]. The morphology and elemental analysis of M-P-S samples were investigated by scanning electron microscopy (SEM) and energy dispersive spectroscopy (EDS) on a Hitachi S3400N system. Samples were prepared by pressing ground powders onto carbon tape on aluminum stubs. EDS samples were prepared in a similar fashion but using thin pellets made with a KBr hand press. TEM was performed using a Hitachi S-7800 transmission electron microscope with an accelerating voltage of 100 kV on sonicated methanol suspended particles. Select electron microprobe analysis was performed on powders on C_wax_ pieces using a JEOL JXA-8230 Electron Probe Microanalyzer. Selected powders on Cwax electrode tips were also examined by EDS elemental mapping after electrocatalysis. Inductively coupled plasma optical emission spectroscopy (ICP-OES, Perkin Elmer Optima DV 7000) was used to obtain quantitative compositional analysis of M-P-S materials dissolved in nitric acid (with heating as needed) versus commercial standards. The vibrational properties of the metal thiophosphate materials were analyzed using KBr pellets on a Nicolet Nexus 760 FT-IR spectrometer. Optical absorption measurements were performed at room temperature using a LabSphere RSA solid-state diffuse reflectance attachment on an HP 8453 UV-Vis spectrometer. The powder samples were embedded on filter paper and placed between two glass microscope slides. The absorption data were converted to Kubelka–Munk units, and the approximate energy band gap and onsets were determined. The band gaps were calculated using the Kubelka–Munk (KM) function [F (R) = (1−R)^2^/2R], where R is the diffuse reflectance of the sample. Extrapolation of the onset absorption events from a plot of the KM function versus energy yields estimated band gap (E_g_) energies. The magnetic susceptibility of the M-P-S materials was analyzed at room temperature (298 K) using ground powders with a Johnson-Matthey MSB magnetic susceptibility balance. Molar magnetic susceptibility (χ_m_) and spin only magnetic moment (μ_B_) values were calculated from mass susceptibilities assuming paramagnetic behavior with correction for core diamagnetism.

### 3.4. Electrochemical Analysis

Working electrodes for electrocatalytic measurements were prepared using graphite/paraffin wax mixture (50% graphite in wax) inside a PTFE tube (C_wax_ electrode) similar to that previously reported by our group [18,66]. This conducting carbon with an adherent (sticky) surface has shown utility in several prior electrochemical studies [76,77]. Working electrode tips were 1.4 cm long, 3.2 mm ID, and 6.4 mm OD, with a 0.080 cm^2^ geometrical surface area and tip ends that were coned to reduce gas bubble adhesion. Prior to catalyst loading, blank C_wax_ electrode tips were connected to brass current collectors and were submerged in a pre-heated water bath (55 °C) for ~15 min. Homogeneous catalyst suspensions were prepared by brief sonication of ~1–2 mg of catalyst with 20 μL of methanol and ~5–10 μL aliquots were placed in an aluminum weigh boat to air dry. The air-dried catalyst-containing aluminum weigh boat was tared in a microbalance and was then placed in a preheated hot plate (55 °C). The softened C_wax_ blank electrode tips were gently pressed onto the catalyst and excess catalyst powder on the PTFE tip lining was carefully returned to the aluminum weigh boat. The electrode tip was re-pressed several times at a clear space in the weigh boat to assure sample powders were firmly embedded on the wax. After the catalyst loading, the aluminum weigh boat was weighed on the previously tared microbalance and the mass of loaded catalyst was recorded; typical catalyst mass loadings ranged from about 0.5 to 1.0 mg.

Electrochemical measurements were performed in a Bioanalytical Systems (BASi) 100b potentiostat using a three-electrode cell with a C_wax_ working electrode, 0.5 M H_2_SO_4_ electrolyte, Hg/Hg_2_Cl_2_ (SCE) reference electrode, and a platinum wire counter electrode; experiments using a graphite rod counter electrode (Alfa Aesar, 6.2 mm diam., SPK grade, 99.9995%) were conducted for comparison. A magnetic cross stir bar was placed directly under the working electrode (~6 mm away) to remove any gas bubbles and to minimize their adherence to the electrode surface. A schematic of the assembled cells and images of the C_wax_ electrode are shown in Appendix A. The SCE electrode potential values were converted to standard hydrogen electrode potentials for different pH values using the *E*_RHE_ = *E*_SCE_ + 0.059pH + *E*^0^_SCE_, with pH = 0.3 (0.5 M H_2_SO_4_) and *E*^0^
_SCE_ = 0.241 V. All reported potentials are referenced to RHE values, and all current densities are calculated using the geometric surface area of the C_wax_ electrode (0.08 cm^2^). The electrolyte solutions were purged with H_2_ gas (ultra-high purity 99.999%, Praxair) that were pre-humidified to minimize electrolyte evaporation by passing it through a water bubbler. Gas purging began 30 min before electrochemical measurements and continued throughout the experiment. HER activities and stability were evaluated using 50 linear sweep voltammograms (LSVs) taken without instrument iR correction. 30 LSV scans were taken with 85% instrument iR correction for comparison (typical R_cell_ values for MPS_3_ materials on C_wax_ tips were M = Co, Ni~60 Ω, and Fe~300 Ω). Care was taken to consider the influence of iR corrections on these catalytic materials as this may mask catalyst charge transfer differences [57,58,59]. The electrochemical surface areas (ECSA) were determined from double-layer capacitance (C_dl_) in the non-Faradaic region using cyclic voltammetry (CV) data at scan rates of 5, 10, 25, 50, and 75 mV/s with H_2_ gas purge [18,58]. The capacitance values were converted to approximate areas using a 35 μF/cm^2^ relationship [78]. The long-term HER stability of MPS_3_ was investigated using 18 h time base chronoamperometry studies (CA) at constant potentials targeting ~10 mA/cm^2^ current density. XRD samples were prepared from the electrodes, post-electrochemical analysis, by cutting ~1–2 mm slices from the end of the C_wax_ electrode containing a thin surface coating of embedded MPS_3_ powders and placing them in the well of an XRD sample holder for XRD analysis and used for EDS elemental mapping as described in our prior work (Appendix A) [66]. The PTFE lining of the electrode tips was removed, and elemental mapping was conducted using electron microprobe analysis.

## 4. Conclusions

The direct solvent-free exchange reaction of anhydrous metal halides with either element P or S, P/S mixtures, or P_2_S_5_/P at 500 °C produces crystalline metal thiophosphates, with P_2_S_5_ reactions generally leading to the phase-pure products, MPS_3_ (M = Fe, Co, Ni) and Cu_3_PS_4_. The MPS_3_ materials contain M^2+^ cations and P_2_S_6_^4−^ anions in a non-metal rich layered structure that form as micrometer-sized and sometimes faceted particles. While these three thiophosphates have similar low energy band gaps (~1.5 eV), they show a great variation in electrocatalytic HER activity when embedded on a conducting wax electrode. Whereas CoPS_3_ shows the highest and most stable HER activity in acidic electrolyte, NiPS_3_ is less active and stable, and FePS_3_ appears highly resistant to performing HER. The effect of potential platinum counter electrode migration is identified for the lower activity NiPS_3_ and FePS_3_ samples. Post-electrocatalytic analysis of particles embedded on the wax electrode show strong evidence of bulk retention of the crystalline MPS_3_ structure.

## Figures and Tables

**Figure 1 molecules-27-05053-f001:**
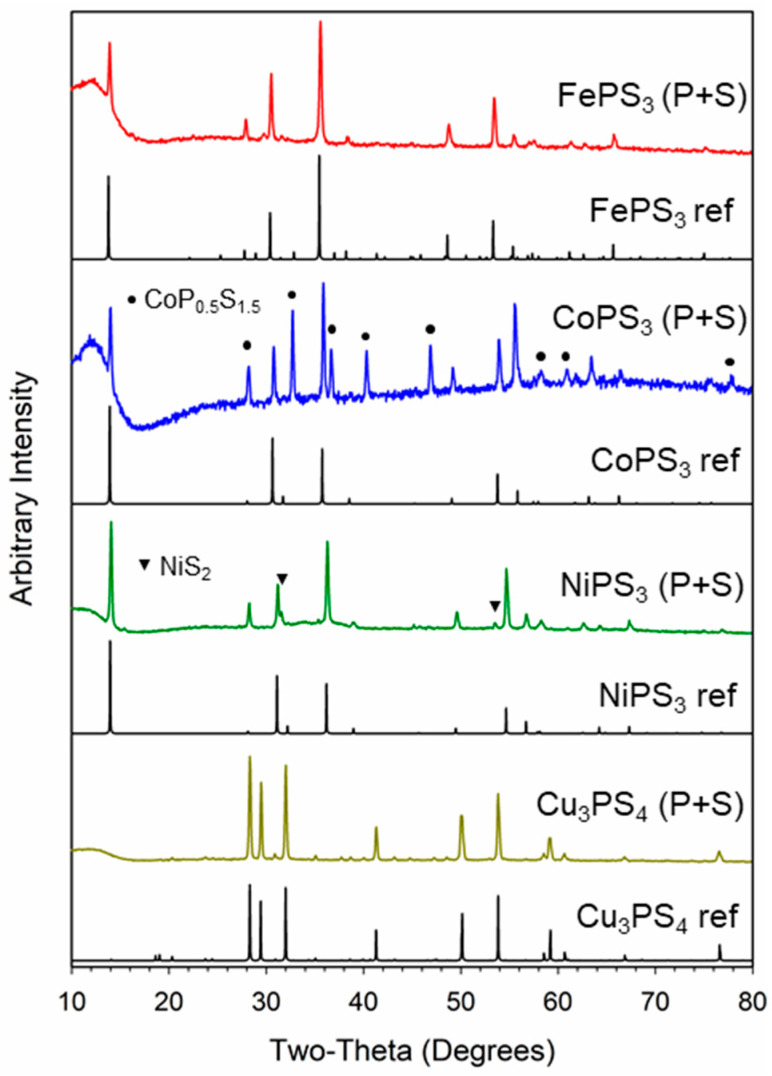
Powder X-ray diffraction results for MPS_3_ (M = Fe, Ni, Co) and Cu_3_PS_4_ products from the reaction of metal halides with elemental sulfur and phosphorus at 500 °C. Reference patterns (ref) are shown in black under each reaction product: Co_0.5_P_0.5_S_1.5_ (●) and NiS_2_ (▼) impurities are identified.

**Figure 2 molecules-27-05053-f002:**
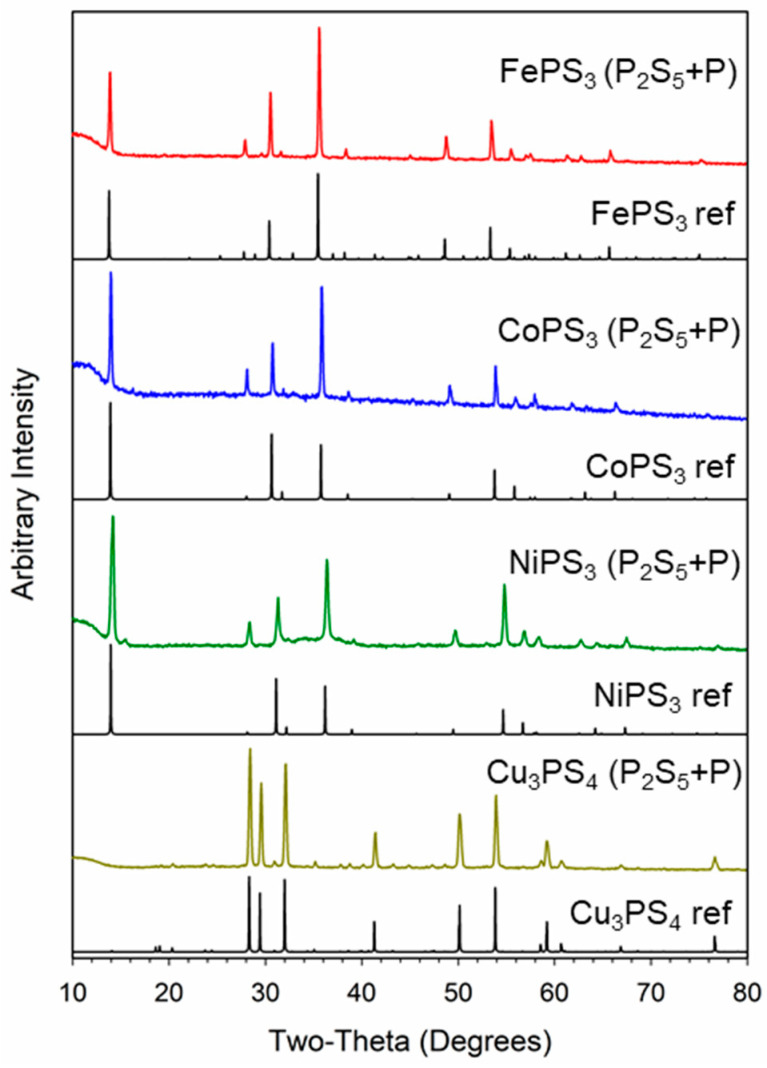
Powder X-ray diffraction results for MPS_3_ (M = Fe, Ni, Co) and Cu_3_PS_4_ products from the reaction of metal halides with molecular P_2_S_5_ and elemental phosphorus at 500 °C. Reference (ref) patterns in black are shown under each reaction product.

**Figure 3 molecules-27-05053-f003:**
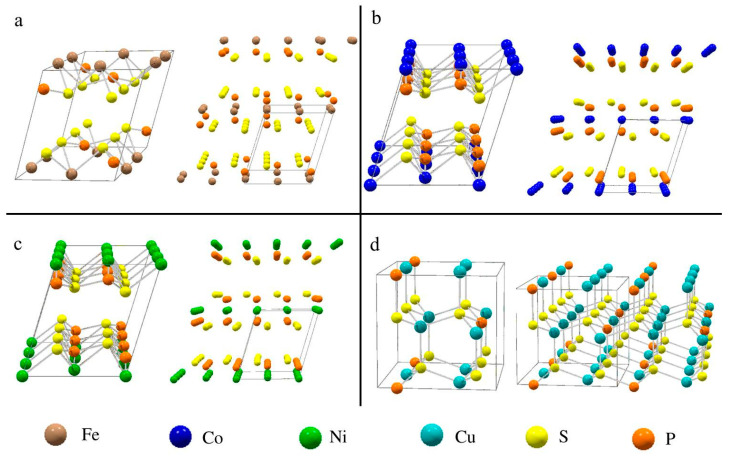
Unit cell and extended structures for the targeted products: (**a**) FePS_3_, (**b**) CoPS_3_, (**c**) NiPS_3_, and (**d**) Cu_3_PS_4_. Extended structures show multiple unit cells.

**Figure 4 molecules-27-05053-f004:**
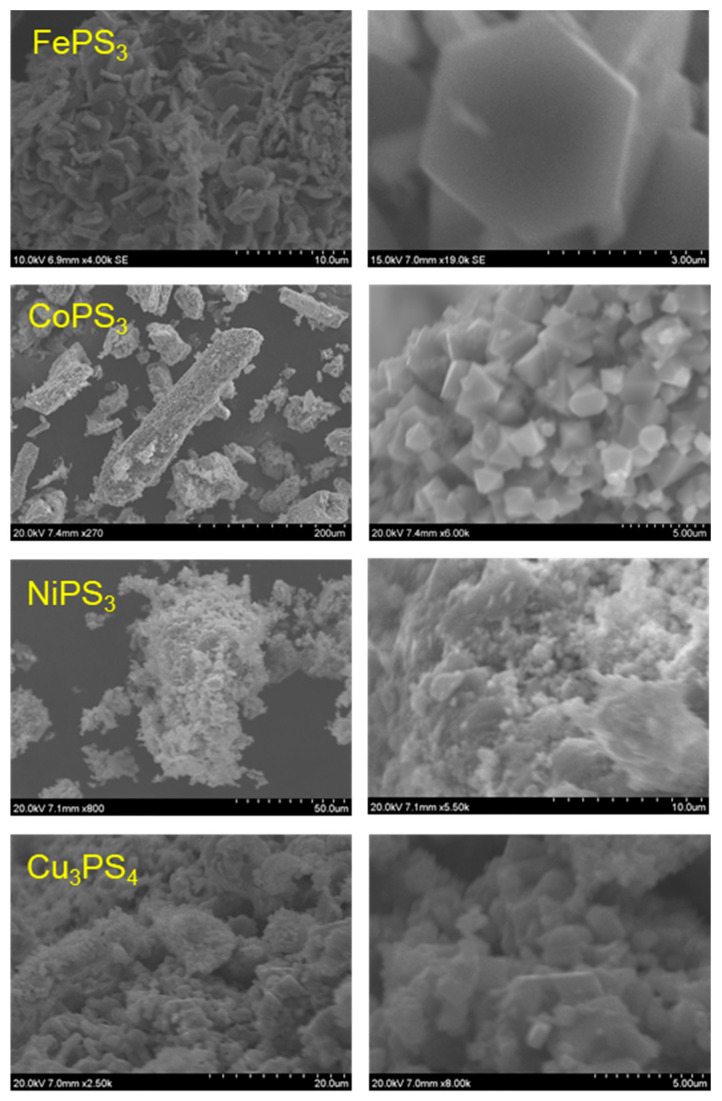
SEM images of M-P-S materials synthesized from metal halides and (P + S) at 500 °C. The right images are higher magnification images of the products shown in the left images.

**Figure 5 molecules-27-05053-f005:**
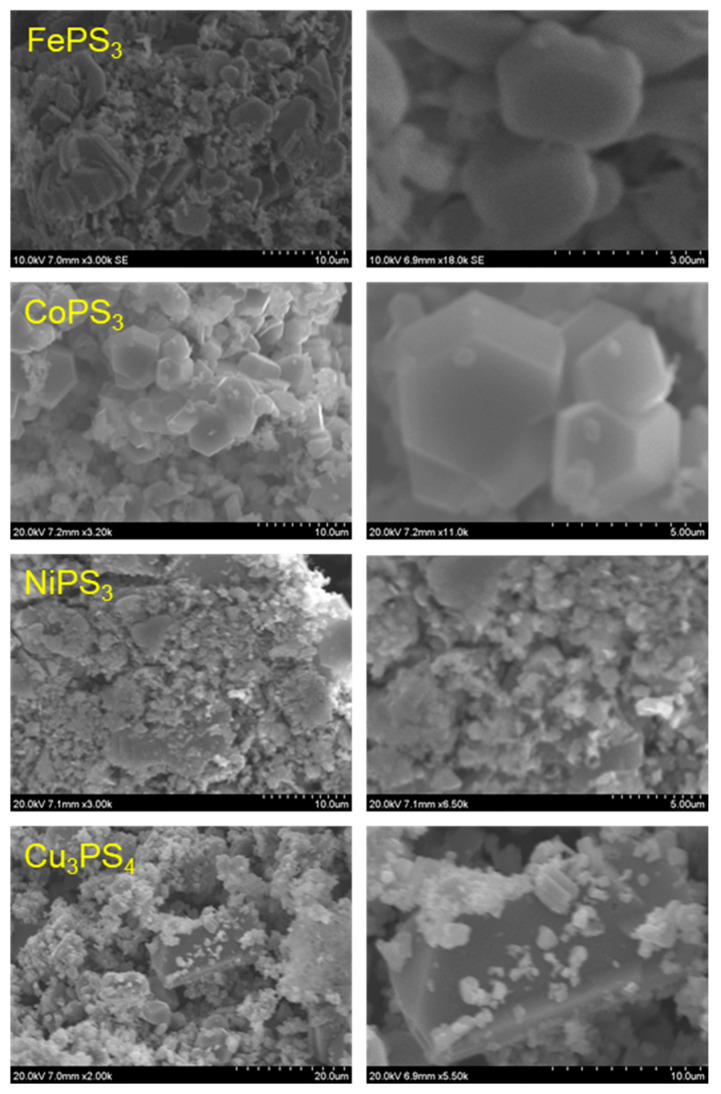
SEM images of M-P-S materials synthesized from metal halides and (P_2_S_5_ + P) at 500 °C. The right images are higher magnification images of the products shown in the left images.

**Figure 6 molecules-27-05053-f006:**
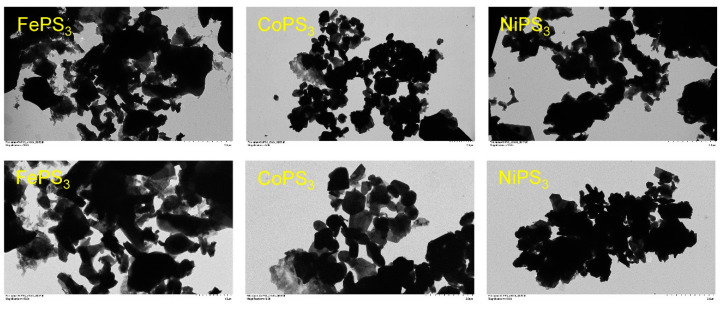
Several representative TEM images of MPS_3_ faceted particles and aggregates synthesized from metal halides and (P_2_S_5_ + P) at 500 °C. All image scale bars are 2 μm in length except bottom left FePS_3_ image is 1 μm long.

**Figure 7 molecules-27-05053-f007:**
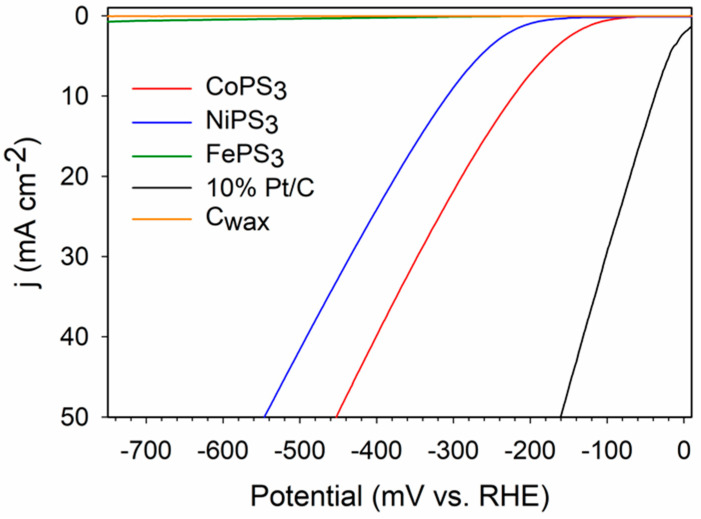
Representative LSV plots for the three MPS_3_ samples (M = Fe, Co, Ni), a C_wax_ blank, and 10%Pt on C powder on C_wax_ electrodes in 0.5 M H_2_SO_4_ (5 mV/s scan rate, with no iR compensation). Data with 85% iR compensation applied are listed in Table 3 and LSV data are in Appendix A.

**Figure 8 molecules-27-05053-f008:**
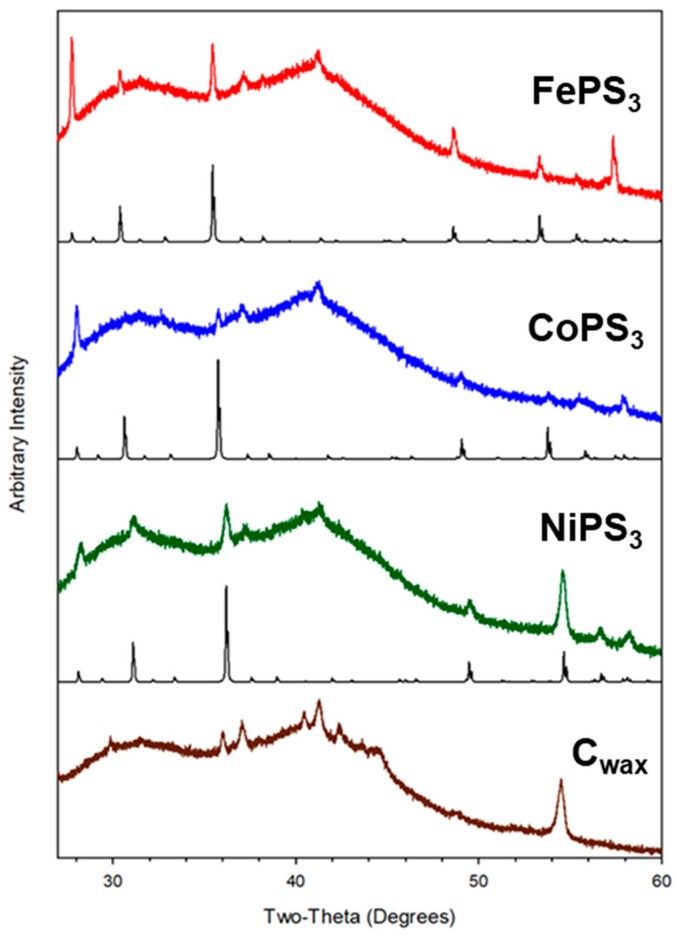
XRD data for MPS_3_ embedded on carbon wax tips after 18 h CA HER experiments. The black standard reference patterns are shown under each data set (FePS_3_-PDF #04-005-1516, CoPS_3_-PDF #01-078-0498, NiPS_3_-PDF #01-078-0499).

**Table 1 molecules-27-05053-t001:** Summary of MPS_3_ reaction product yield, phase, and composition.

Reaction	Target	Yield ^1^	XRD Phases ^2^	M:P:S:Cl Atomic Ratio (EDS)	M:P:S Atomic Ratio (ICP)
FeCl_3_ (P + S)	FePS_3_	93%	FePS_3_	1:1.29:3.04:0.08	1:0.94:3.01
FeCl_3_ (P_2_S_5_ + P)	FePS_3_	84%	FePS_3_	1:1.36:3.24:<0.01	1:0.99:3.58
CoCl_2_ (P + S)	CoPS_3_	62%	**CoPS_3_**, CoP_0.5_S_1.5_	1:1.95:4.52:<0.01	1:0.40:1.62
CoCl_2_ (P_2_S_5_ + P)	CoPS_3_	79%	CoPS_3_	1:1.97:5.13:<0.01	1:0.73:2.42
NiCl_2_ (P + S)	NiPS_3_	63%	**NiPS_3_**, NiS_2_	1:1.34:3.34:0.02	1:0.83:4.13
NiCl_2_ (P_2_S_5_ + P)	NiPS_3_	87%	NiPS_3_	1:1.24:3.20:<0.01	1:0.79:4.23

(1) Mass yield based on theoretical yield of targeted phase. (2) XRD bolded phases are major phases in multiphase systems.

**Table 2 molecules-27-05053-t002:** Comparison of the standard (298 K) heats of formation and the calculated eV/atom energies for several Fe, Co and Ni phosphides, sulfides, and MPS_3_ materials.

	**MP_2_**	**MS_2_**	**MPS**	**MPS_3_**
	**FeP_2_**	**FeS_2_**	**FePS**	**FePS_3_**
**ΔH_f_ (kJ/mol)**	−221	−172	*n*/a	*n*/a
**ΔH_f_ (eV/atom)**	−0.537	**−0.948**	−0.754	−0.648
	**CoP_3_**	**CoS_2_**	**CoPS**	**CoPS_3_**
**ΔH_f_ (kJ/mol)**	−280	−153	*n*/a	*n*/a
**ΔH_f_ (eV/atom)**	−0.497	**−0.774**	−0.773	−0.613
	**NiP_2_**	**NiS_2_**	**NiPS**	**NiPS_3_**
**ΔH_f_ (kJ/mol)**	−129	−134	*n*/a	*n*/a
**ΔH_f_ (eV/atom)**	−0.361	**−0.684**	−0.575	−0.587

*n*/a = not available.

**Table 3 molecules-27-05053-t003:** Summary of key electrochemical and electrocatalytic HER results for MPS_3_ materials ^1^.

Catalyst Material	Onset (mV) ^2^	10 mA/cm^2^ (mV)	20 mA/cm^2^ (mV)	Tafel Slope (mV/dec) ^3^	ECSA (cm^2^) ^4^	Extended Stability ^5^
**CoPS_3_**	−96 ± 1 (−80 ± 1)	−222 ± 2 (−169 ± 1)	−289 ± 4 (−198 ± 2)	−71 ± 5	6/2	87% @ −241 mV
**NiPS_3_**	−178 ± 9 (−174 ± 2)	−311 ± 8 (−261 ± 2)	−378 ± 10 (−290 ± 3)	−86 ± 4	2/1	84% @ −401 mV
**FePS_3_**	~500 (~270)	*n*/a	*n*/a	*n*/a	2/1	29% @ −749 mV
**10% Pt on C**	47 ± 9 (122 ± 2)	−31 ± 4 (−8 ± 1)	−57 ± 8 (−33 ± 2)	−49 ± 2	27/44	22% @ −79 mV

(1) LSV results for powders embedded on 0.08 cm^2^ carbon-wax electrode in 0.5 M H_2_SO_4_, graphite counter, SCE reference, and mV converted to RHE. Average of LSV data from 50 LSV runs with no iR compensation (in parentheses are results for 30 LSV runs with 85% iR compensation). (2) Estimated potential required for 0.5 mA/cm^2^ current density. (3) Calculated using linear region near 2 mA/cm^2^ and averaging results from 50 LSVs. (4) Values obtained before/after 50 LSV runs without iR compensation. (5) Percent change from 15 min to 18 h mark for constant potential experiment.

## Data Availability

Experimental data used for graphical results shown in this study are available upon request from the corresponding author. Appendix A provided with this paper contains additional tabular and graphical data that supports the reported results.

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
