# Peer review of "Facile Solvent-Free Synthesis of Metal Thiophosphates and Their Examination as Hydrogen Evolution Electrocatalysts"

_molecules, 2022, doi:10.3390/molecules27165053_

Round 1
Reviewer 1 Report
Ms. Ref. No.: Molecules-1815512
The manuscript entitled "Facile Solvent-Free Synthesis of Metal Thiophosphates and Their Examination as Hydrogen Evolution Electrocatalysts" was studied.
Some comments should be addressed in manuscript to fulfil the requirement of a proper manuscript.
1) This work is not as novel as it said. As mentioned lots of works previously worked on this subject.
2) Authors talked about the morphology of the catalyst, please give explanation about how the morphology affected HER results.
3) Line 369-371- What does this statement mean “Because NiPS3 and FePS3 evolve hydrogen at more negative potentials, platinum may be deposited on these materials at longer times when a Pt CE is used”?
Author Response
Reviewer 1
The manuscript entitled "Facile Solvent-Free Synthesis of Metal Thiophosphates and Their Examination as Hydrogen Evolution Electrocatalysts" was studied. Some comments should be addressed in manuscript to fulfil the requirement of a proper manuscript.
Reviewer 1 comment 1: This work is not as novel as it said. As mentioned lots of works previously worked on this subject.
Our response: This is true regarding both MPS3 syntheses and their HER activity as we describe in lines 59-85 of the introduction. This paper’s novelty lies mainly in the solvent-free synthetic approach and making HER measurements with well-crystallized, often micrometer sized MPS3 samples. We modified and better clarified parts of the introduction text to highlight motivation for this work in the context of prior studies.
Reviewer 1 comment 2: Authors talked about the morphology of the catalyst, please give explanation about how the morphology affected HER results.
Our response: FePS3 and CoPS3 show the largest, faceted crystallites, while NiPS3 grows as more aggregated particles (Figure 5). It is unclear if there is a direct relationship between particle size and shape as it relates to HER activity. In response to another reviewer comment, we have added TEM images for several MPS3 samples examined for HER (new Figure 6) that show similarly faceted groups of aggregated microparticles. We have added some text discussion on page 14 on relationships between SEM/TEM and ECSA and particle size faceting.
Reviewer 1 comment 3: Line 369-371- What does this statement mean “Because NiPS3 and FePS3 evolve hydrogen at more negative potentials, platinum may be deposited on these materials at longer times when a Pt CE is used”?
Our response: We see that this sentence is unclear. We reworded it to emphasize that the higher negative applied potentials for FePS3 and NiPS3 that are used for long 18-hour CA experiments may be sufficient to reduce platinum ions from solution onto the catalyst surface when a Pt CE is used. This would improve the catalyst HER activity over time.
Reviewer 2 Report
According to this study, Nathaniel Coleman Jr.et al., used a solvent-free method to synthesize several metal phosphonothioates. Simultaneous, the hydrogen evolution reaction and the trends of MPS3 materials as electrocatalysts under acidic conditions was also investigated. Unfortunately, I cannot recommend this manuscript for publication because the following reasons:
1. Significant structural characterization, such as transmission electron microscopy, is missing from the article.
2. Important electronic structure characterization, such as XPS, is missing from the article.
3. Figure S4B tested the performance at different potentials, but there is too much data to distinguish between them.
4. The HER performance of the catalyst under acidic conditions is too poor to be of practical use.
5. The Tafel slope and ECSA data appear in Table 3, but the original fit plots do not appear in the article or in the supporting information.
6. The authors should further investigate the effect of different temperatures and the absence of phosphorus/sulphur content on the performance and structure of the catalyst.
7. The colors of the microprobe elemental mapping in Figure S7A and Figure S7B are hard to be distinguished.
Author Response
Reviewer 2
Reviewer 2: According to this study, Nathaniel Coleman Jr.et al., used a solvent-free method to synthesize several metal phosphonothioates. Simultaneous, the hydrogen evolution reaction and the trends of MPS3 materials as electrocatalysts under acidic conditions was also investigated. Unfortunately, I cannot recommend this manuscript for publication because the following reasons:
Our response: We have gone through the entire paper to improve the clarity of our discussion, improve English and grammar in several places, and to expand on items that may be unclear (also in response to other reviewer comments). We hope our revisions and clarifications have improved this submission for publication consideration.
Reviewer 2 Comment 1: Significant structural characterization, such as transmission electron microscopy, is missing from the article.
Our response: The MPS3 materials synthesized by our solvent-free route are well-known and we have characterized these generally large crystallites by SEM, XRD, bulk chemical analysis, IR, optical and magnetic measurements. We have obtained TEM images for the sample’s examined by HER and include those in new Figure 6. They show similar small faceted microparticles that are highly aggregated.
Reviewer 2 Comment 2: Important electronic structure characterization, such as XPS, is missing from the article.
Our response: The band gaps of these MPS3 materials are well-known and our measurements verified the ~1.5 eV band gaps for these solids. We also have bulk elemental analysis data for these crystalline solids, so surface sensitive XPS would not provide substantial additional structural/bonding information on the bulk solid. It is very likely that the surface will contain surface oxide/hydroxide bonds with metal and non-metal components, as we have seen in other work. We considered XPS for the post-HER catalyst particles on electrode surfaces (in addition to XRD and SEM data presented in this submission, Figures 7 and S8), but unfortunately our XPS system has been under extended maintenance for nearly 6 months due to parts shortages that seem pandemic related (the company is unable to provide replacements, so in-house repairs are underway). Thus, we are unable to obtain surface structural or valence band info on these MPS3 materials. This has been highly frustrating for us to better analyze surface structures of materials.
Reviewer 2 Comment 3: Figure S4B tested the performance at different potentials, but there is too much data to distinguish between them.
Our response: We have clarified the meaning and source of the bottom run number (x-axis) plots shown in new Figures S3 – S5. These are created by taking mV and current density data from the LSV overlay plots at top of each figure page and plotting them versus run number to show how LSVs are changing with each LSV scan (stable or moving to higher mV or lower mV values to achieve a specific current density). Without these extracted data plots, we agree that it is difficult to see trends in the 50 LSV overlay data. In particular, the S4B plot shows that LSVs move towards higher applied potentials with each LSV run, but still only show very low to no HER activity (~1 mA/cm2 at -350 mV). We have modified the figure captions to better clarify how these graphs were generated from the LSV overlay plots.
Reviewer 2 Comment 4: The HER performance of the catalyst under acidic conditions is too poor to be of practical use.
Our response: We agree that the iron sample data shown in Figure S4B has negligible HER activity but the CoPS3 with its very large crystallites shows moderate HER activity (~-170 mV applied potentials for 10 mA/cm2 after iR correction in Figure S2B) and shows good 18 hr constant potential current densities. We agree that these MPS3 materials are unlikely to be of practical or industrial use in an electrolyzer, but this synthetic method shows some useful solvent-free flexibility and MPS3 materials or related solids produced by these methods may find use in other electrocatalytic or photocatalytic processes. For example, MPS3 materials may potentially be grown on other solids via this method to produce composite structures.
Reviewer 2 Comment 5: The Tafel slope and ECSA data appear in Table 3, but the original fit plots do not appear in the article or in the supporting information.
Our response: We apologize for this oversight and clarity issue. We have added representative Tafel plots and scan rate graphs used for the ECSA calculations into supporting information (new Fig. S6-7). The Tafel slopes in Table 3 are an average of results from 50 LSVs that overlay in semi-log plots, so we show one representative graph (standard deviations in Table 3 are small).
Reviewer 2 Comment 6: The authors should further investigate the effect of different temperatures and the absence of phosphorus/sulphur content on the performance and structure of the catalyst.
Our response: With this synthetic study, we intended to make it comparable to our prior MP2/MP3 studies that kept reactions at or below 500 °C. Instead of a more in-depth reaction temperature dependence study, we have focused on the different P/S reactant strategies. It is certainly possible/likely that lowering reaction temperature may lead to smaller crystallites while higher temperatures and variations in P/S stoichiometries may lead to different phases such as the MPS structures as we did identify CoPS in some products. As noted in the SI info, if phosphorus is not in the reaction ampoule, then no reaction occurs with just sulfur and MCl2. These are worthy of future experiments, but beyond the scope of our initial study in this area.
Reviewer 2 Comment 7: The colors of the microprobe elemental mapping in Figure S7A and Figure S7B are hard to be distinguished.
Our response: We agree that the microprobe surface elemental images are not sufficiently clear (this instrument plots data using a heat map style of color contrast). Repair work on our EDS system necessitated use of the microprobe. We added text to the microprobe SI graphs to clarify the color scheme. We analyzed our electrode tip catalyst samples (after CA using graphite CE) with EDS and provide some similar but better color contrast elemental maps (Figures S12 A-C).
Reviewer 3 Report
The manuscript under review presents new synthetic routes for the preparation of catalysts for hydrogen evolution reaction based on metal thiophosphates. The obtained materials are characterized using SEM and XRD. The manuscript might be published; however, the authors should solve some issues before possible publication.
1. Unfortunately, the authors HER present polarization curves with no IR compensation (Fig. 6). This makes it difficult to compare the obtained results with the data of other authors. The electrical conductivity of prepared materials is unknown. The authors do not report anything about the thickness of the catalyst layer on the electrode surface.
2. It is not clear how the kinetic parameters of HER were calculated (Table 3). According to th authors data, the Tafel slope for CoPS3 should be (–289+222)/(log 20 – log 10) = –222 mV dec–1, not –71 mV dec–1. The authors have to provide a semi-log plot for HER.
3. The data on the stability of the material during long-term electrolysis are highly desirable.
Author Response
Reviewer 3
Reviewer 3: The manuscript under review presents new synthetic routes for the preparation of catalysts for hydrogen evolution reaction based on metal thiophosphates. The obtained materials are characterized using SEM and XRD. The manuscript might be published; however, the authors should solve some issues before possible publication.
Our response: We have also characterized these materials by ICP bulk elemental analysis and have added TEM images requested by another reviewer. We are hopeful that the clarifications, additional information, and clarifications below make this work worthy of publication.
Reviewer 3 Comment 1: Unfortunately, the authors HER present polarization curves with no IR compensation (Fig. 6). This makes it difficult to compare the obtained results with the data of other authors. The electrical conductivity of prepared materials is unknown. The authors do not report anything about the thickness of the catalyst layer on the electrode surface.
Our response: We apologize for this clarity issue. In accordance with best practices described in several recent literature reviews, we have reported both iR uncompensated and 85% IR compensated data. We have modified the new Figure 7 caption to point the reader to the IR corrected data. We also added an LSV overlay graph in Figure S2. We added Rcell values in the 85% IR compensated SI figure captions. These electrode surfaces contain particles embedded on the surface. We used bulk densities, electrode diameter, and catalyst amount to estimate thickness of powder on the surface (~ 0.04 mm for 1 mg sample @ 3 g/cm3 density) and described this estimated thickness on page 12 of the manuscript.
Reviewer 3 Comment 2: It is not clear how the kinetic parameters of HER were calculated (Table 3). According to th authors data, the Tafel slope for CoPS3 should be (–289+222)/(log 20 – log 10) = –222 mV dec–1, not –71 mV dec–1. The authors have to provide a semi-log plot for HER.
Our response: We have added representative Tafel analysis graphs to Figure S6 and clarified in the text that calculated Tafel slopes for regions of low current density (much lower than 10 mA/cm2) to avoid issues with high H2 formation that can introduce mass transport issues. Current density regions near LSV onsets ~2 mA/cm2 showed linear regions on the semi-log plots were used to estimate the Tafel slopes (taken as an average of multiple LSV curves, standard deviations in Table 3).
Reviewer 3 Comment 3: The data on the stability of the material during long-term electrolysis are highly desirable.
Our response: We provide a summary of 18-hour constant applied potential chronoamperometry (CA) experiments in Table 3 and show the current density versus time graphs in Figure S5 and S6. Figure 7 shows the retention of crystalline structure after the CA experiments and EDS elemental maps on catalyst is given in supporting information (New Figure S12). We did not subject these MPS3 materials to longer term electrolysis experiments, but the CoPS3 appears best suited to examine for that. This is beyond the scope of our synthetically focused paper with its HER data to compare with prior studies on these MPS3 materials.
Reviewer 4 Report
This paper reports on the novel procedure for the synthesis of crystalline metal thiophosphates, and their detailed study including the study of crystal structure, chemical composition, microstructure, optical and magnetic characteristics, and thermochemical properties. Furthermore, electrocatalytic properties of the materials obtained are also provided on an example of hydrogen evolution reactions. The information presented is quite new and the paper definitely deserves to be published in Molecules as received. This is a good advance in the field of inorganic chemistry.
Author Response
This paper reports on the novel procedure for the synthesis of crystalline metal thiophosphates, and their detailed study including the study of crystal structure, chemical composition, microstructure, optical and magnetic characteristics, and thermochemical properties. Furthermore, electrocatalytic properties of the materials obtained are also provided on an example of hydrogen evolution reactions. The information presented is quite new and the paper definitely deserves to be published in Molecules as received. This is a good advance in the field of inorganic chemistry.
Our response: We are grateful for the positive review of our submission. We hope that with revisions requested by the other reviewers, that this work is even stronger than it was before.
Round 2
Reviewer 2 Report
The manuscript were greatly improved, and the current manuscript can be accepted.
Reviewer 3 Report
The manuscript can be published.